# Maximum entanglement of mixed symmetric states under unitary transformations

**Eduardo Serrano-Ensástiga[1,2]⋆ and John Martin[1]†**

**1** Institut de Physique Nucléaire, Atomique et de Spectroscopie, CESAM,
University of Liège, B-4000 Liège, Belgium
**2** Centro de Nanociencias y Nanotecnología, Universidad Nacional Autónoma de México,
Apdo. Postal 14, 22800 Ensenada, Baja California, México

⋆ ed.ensastiga@uliege.be , † jmartin@uliege.be

## Abstract

We study the maximum entanglement that can be produced by a global unitary transformation for systems of two and three qubits constrained to the fully symmetric states. This restriction to the symmetric subspace appears naturally in the context of bosonic or collective spin systems. We also study the symmetric states that remain separable after any global unitary transformation, called symmetric absolutely separable (SAS) states, or absolutely classical for spin states. The results for the two-qubit system are deduced analytically. In particular, we determine the maximal radius of a ball of SAS states around the maximally mixed state in the symmetric sector, and the minimal radius of a ball that contains the set of SAS states. As an application of our results, we also analyse the temperature dependence of the maximum entanglement that can be obtained from the thermal state of a spin-1 system with a spin-squeezing Hamiltonian. For the symmetric three-qubit case, our results are mostly numerical, and we conjecture a 3-parameter family of states that achieves the maximum negativity in the unitary orbit of any mixed state. In addition, we derive upper bounds, apparently tight, on the radii of balls containing only/all SAS states.

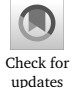

# 1 Introduction and problem statement

Entanglement is both a fundamental concept of quantum theory and a central resource of quantum technology applications, ranging from quantum communication and cryptography, quantum sensing and metrology, quantum simulation to quantum computing [1–6]. In a multipartite quantum system, entanglement can be created by applying an appropriate unitary transformation on a pure product state. This transformation cannot be *local*, as local unitary operations cannot change the entanglement content of a state. *Global* unitary transformations, on the other hand, have the potential to increase the entanglement among the parties [7,8]. They can be implemented from the unitary time evolution under a Hamiltonian describing e.g. interactions among the subsystems or between the subsystems and external driving fields or a tailored experimental device [9,10]. As quantum mechanics is time reversible, this entanglement can also be removed by applying the inverse unitary transformation.

Although it is at the heart of many protocols leading to a quantum advantage, entanglement remains one of the most delicate quantum properties to preserve from unwanted interactions with the environment. When a system interacts with its surrounding, its state must be described by a density operator $\rho \in \mathcal{B}(\mathcal{H})$, where $\mathcal{B}(\mathcal{H})$ is the set of the bounded linear operators acting on the Hilbert space of the quantum states $\mathcal{H}$. System-environment interactions generally tend to deteriorate the coherence and decrease the entanglement content of a state. After a sufficiently long decoherence time, an initially entangled state may lose all its entanglement and become a mixed separable state $\rho_{\text{sep}}$. This can even reach a point where no *global* unitary transformation applied on $\rho_{\text{sep}}$ is capable of creating entanglement. The state is then said to be Absolutely Separable (AS) [11]. For a given system, it is obviously of interest to know which states are absolutely separable, as these states are of little or no use for applications that require entanglement. More generally, it is important to know what is the maximum amount of entanglement that can be obtained from a mixed state by the sole application of unitary transformations. To answer this question, it is first necessary to choose a measure of entanglement [6], i.e., a scalar function $E(\rho)$ of quantum states that satisfies a series of conditions [6] such as $E(\rho) = 0$ if and only if $\rho$ is separable. In this work, we will only deal with quantum states which can be represented by a two-qubit or a qubit-qutrit system – cases for which the Positive Partial Transpose (PPT) criterion is a necessary and sufficient condition for

entanglement [12–14] – and use as a measure of entanglement the negativity, defined by

$$\mathcal{N}(\rho) \equiv \frac{\|\rho^{T_A}\|_1 - 1}{2} = \sum_k \frac{|\Lambda_k| - \Lambda_k}{2}, \tag{1}$$

with $\Lambda_k$ the eigenvalues of the partial transpose of $\rho$ with respect to a subsystem $A$, $\rho^{T_A}$, and $\|X\|_1 \equiv \mathrm{Tr}\sqrt{X^\dagger X}$ the trace norm. Equipped with this entanglement measure, the aim is then to find, for any $\rho$, the state in its global unitary orbit, $\{U\rho U^\dagger : U \in SU(N)\}$ where $N = \dim(\mathcal{H})$, that maximizes the negativity. This maximum will depend only on the eigenspectrum of $\rho$ because these are the only invariant quantities on its $SU(N)$ orbit.

The emblematic case of a bipartite quantum system composed of two qubits, with Hilbert space $\mathcal{H} \simeq \mathbb{C}^2 \otimes \mathbb{C}^2 \simeq \mathbb{C}^4$, was solved in a seminal paper by Verstraete, Audenaert, and De Moor [15]. They showed that the maximum negativity achieved in the $SU(4)$ orbit of a state $\rho$ with eigenvalues sorted in nonascending order $\lambda_1 \geq \lambda_2 \geq \lambda_3 \geq \lambda_4$ is

$$\max_{U \in SU(4)} \mathcal{N}\left(U\rho U^\dagger\right) = \max\left(0, \sqrt{(\lambda_1 - \lambda_3)^2 + (\lambda_2 - \lambda_4)^2} - \lambda_2 - \lambda_4\right). \tag{2}$$

In particular, a complete characterisation of the set of AS two-qubit states follows from setting the previous equation to zero. Other aspects concerning the set of AS states for bipartite systems have been discussed in the literature. To mention a few, it has been shown that AS states form a convex and compact set, and that one can construct operators to witness absolute separability [16] or lack thereof [17]. Recently, quantum maps that output absolutely separable states have been analysed in Ref. [8]. Similar questions on absolute versions of basic quantum properties over (global) unitary orbits have also been studied, such as for PPT [18], locality [19], unsteerability [20], non-negative conditional entropy [21], or quantum discord, see Ref. [22] for an overview. Another variation of the problem is the calculation of the maximum entanglement achievable in the set of quantum states with fixed purity [23, 24], which is a larger set than the $SU(N)$ orbit of a state. The negativity stands as an entanglement measure for the qubit-qutrit system, because of the PPT criterion. However, the question mentioned above for this system remains open except for particular states [25, 26].

In some cases, physical constraints impose a restriction on the set of unitary transformations that can be applied to a state [27, 28]. For instance, in systems of identical and indistinguishable bosons, such as photons, an $N$-qubit state $\rho$ has to be invariant under any pair of permutation matrices, $\pi_\sigma$, i.e. $\rho = \pi_\sigma \rho \pi_{\sigma'}$. Another example is a spin-$s$ system which is, either physically or conceptually, equivalent to a $2s$ symmetric multiqubit system. For these systems, the set of physical states is reduced to the symmetric subspace $\vee^N \mathbb{C}^2 \equiv (\mathbb{C}^2)^{\vee N} \subset \otimes^N \mathbb{C}^2 \equiv (\mathbb{C}^2)^{\otimes N}$ of dimension $N + 1$, and hence the global unitary transformations are limited to $SU(N + 1)$ linear operations within this subspace. The subsystems of indistinguishable multipartite systems can also carry entanglement, although there is a debate in the literature suggesting that it might be artificial due to exchange symmetry [29]. Nevertheless, entanglement in indistinguishable systems is a relevant resource for some applications in quantum metrology and quantum information with bosonic systems, see e.g. [30]. The questions posed above arise naturally in this context, such as what is the maximum amount of entanglement in the $SU(N + 1)$ orbit of a symmetric mixed state. Even for the simplest case of two qubits, this question has not been answered. The main objective of this work is to fill this gap for symmetric two- and three-qubit systems. For these symmetric systems, negativity is also a proper measure of entanglement because they are subspaces of qubit-qubit and qubit-qutrit systems, respectively. However, since the allowed unitary operations are restricted, the maximum negativity of a symmetric state will, in general, not be equal to the unrestricted case, as we show below. The differences between the two problems can be examined more closely

for two qubits. On the one hand, the state $\rho_S$ of the system must belong to the symmetric sector $(\mathbb{C}^2)^{\vee 2}$ of $(\mathbb{C}^2)^{\otimes 2}$, of dimension 3. On the other hand, the global unitary operations $SU(4)$ are now restricted to the subset of operations that leave the symmetric sector $(\mathbb{C}^2)^{\vee 2}$ invariant, which is equivalent to $SU(3)$. The latter modification considerably changes the orbit of the state $\rho_S$ and, consequently, the maximally entangled state in its $SU(3)$ orbit is different from the one attained in its full $SU(4)$ orbit.

The analogue of AS states in bosonic systems are those that remain separable after any unitary transformation preserving the fully symmetric subspace. We will call them symmetric absolutely separable (SAS) states (called absolutely symmetric separable states in Ref. [31]), and we will denote the whole set by $\mathcal{A}_{\text{sym}}$. The existence of balls of SAS states around the maximally mixed state in the symmetric sector was shown in Ref. [32]. Similar balls of AS states in the full Hilbert space have been analysed for qubit-qudit systems [11, 33–35], and for the implementation of quantum computation in NMR experiments [36]. In the language of spin states, the SAS states are the equivalents of the Absolutely Classical (AC) spin states introduced in Ref. [32], see Sec. 2 for more details on the correspondence.

The present work is organised as follows: Sec. 2 reviews the definition of separability, classicality of spin states, and their absolute versions over global unitary operations. In Secs. 3 and 4, we calculate the maximum entanglement achieved in the unitary orbits of symmetric two- and three-qubit states, the first system studied analytically while the second one mostly numerically, respectively. For these systems, we then determine the maximal radius of balls contained in $\mathcal{A}_{\text{sym}}$ and the minimal ball that includes $\mathcal{A}_{\text{sym}}$, both around the maximally mixed state in the symmetric sector. In particular, for the two-qubit case, we study the maximum negativity and its temperature dependence for the Lipkin-Meshkov-Glick model [37, 38]. We present the conclusions of this work and some perspectives in Sec. 5.

## 2 Separability and classicality

### 2.1 Separable states of multiqubit systems

The Hilbert space $\mathcal{H}_1$ of a single qubit system is spanned by two basis vectors $|+\rangle$ and $|-\rangle$. The full Hilbert space of an $N$-qubit system $\otimes^N \mathcal{H}_1 \equiv \mathcal{H}_1^{\otimes N}$ is of dimension $2^N$ and is spanned by the product states $|\psi_1\rangle \otimes \cdots \otimes |\psi_N\rangle$ with $|\psi_k\rangle \in \{|+\rangle, |-\rangle\}$ for all $k = 1, \dots, N$. The convex hull of the product states defines the set of *separable* states $\mathcal{S} \subset \mathcal{B}(\mathcal{H}_1^{\otimes N})$. Any state $\rho$ that is not separable, i.e. $\rho \notin \mathcal{S}$, is said to be entangled. All separable states $\rho_{\text{sep}} \in \mathcal{S}$ have zero negativity, $\mathcal{N}(\rho_{\text{sep}}) = 0$. The measure of entanglement of a state cannot, by definition, be modified by local unitary operations [6]. On the other hand, the entanglement of a state $\rho$ may change under a global unitary operation $U \in SU(2^N)$. However, there are special states that remain separable for all $U \in SU(2^N)$ and these are called *absolutely separable* (AS) states [11]. They can be formally defined as the states $\rho \in \mathcal{B}(\mathcal{H}_1^{\otimes N})$ for which

$$\max_{U \in SU(2^N)} E(U \rho U^\dagger) = 0, \tag{3}$$

for some measure of entanglement $E$.

### 2.2 Separable states in the symmetric sector and classical spin states

A multiqubit system is equivalent to a system of $N$ spin-$1/2$, where each of the spin Hilbert spaces are spanned by the eigenvectors of the angular momentum operator $S_z$, the $|1/2, \pm 1/2\rangle$ states that we can identify with the $|\pm\rangle$ qubit states. For our purposes, we only consider the symmetric sector $\vee^N \mathcal{H}_1 \equiv \mathcal{H}_1^{\vee N}$ of $\mathcal{H}_1^{\otimes N}$, spanned by the symmetric Dicke states $|D_N^{(k)}\rangle$ [39]

with

$$|D_N^{(k)}\rangle = K \sum_\pi \pi\Big(\underbrace{|+\rangle \otimes \dots |+\rangle}_{N-k} \otimes \underbrace{|-\rangle \otimes \dots |-\rangle}_{k}\Big), \quad \text{for } k = 0, \dots N, \tag{4}$$

where the sum runs over all the permutations $\pi$ of the qubits and $K > 0$ is a normalization constant. Due to the fact that the $|D_N^{(k)}\rangle$ states are equivalent to the eigenvectors $|s, m\rangle$ of the collective operator $S_z$, with $s = N/2$ and $m = (N-2k)/2$ [39], the subspace $\mathcal{H}_1^{\vee N}$ is isomorphic to the Hilbert space $\mathcal{H}^{(s)}$ of a spin $s$ system, both being of dimension $N + 1 = 2s + 1$. Global unitary transformations restricted in $\mathcal{H}_1^{\vee N}$ correspond to $SU(N+1)$ transformations in $\mathcal{H}^{(s)}$.

The restriction of product states to the symmetric subspace leads to $N$-qubit states of the form $|\psi\rangle = |\phi\rangle^{\otimes N}$ with $|\phi\rangle = \alpha|+\rangle + \beta|-\rangle$ a normalized single qubit state. In the spin picture, this corresponds to *spin-coherent* (SC) states [40–42]. The convex hull of SC states defines the set of *classical* spin-states $\mathcal{C}$ [43,44]. A spin-$s$ state $\rho^{(s)}$ is called *absolutely classical* (AC) when the $SU(2s+1)$ orbit of $\rho^{(s)} \in \mathcal{B}(\mathcal{H}^{(s)})$ is contained in $\mathcal{C}$ [32]. The complement of the set of classical states has also been studied in the literature [43–47] and a measure of non-classicality, called quantumness, has been defined in [45] as the distance between a state $\rho^{(s)}$ and $\mathcal{C}$ [46] (see also [48] for the relation between quantumness and the geometric measure of entanglement).

Now, we introduce formally the notion of *symmetric absolutely separable* (SAS) states, the set of which will be denoted by $\mathcal{A}_{\text{sym}}$. We say that $\rho_S \in \mathcal{A}_{\text{sym}}$ if its $SU(N+1)$ orbit,

$$\{U_S \rho_S U_S^\dagger : U_S \in SU(N+1)\}, \tag{5}$$

contains only separable symmetric states. Equivalently, $\rho_S$ is SAS if

$$\max_{U_S \in SU(N+1)} E(U_S \rho_S U_S^\dagger) = 0, \tag{6}$$

for some measure of entanglement $E$. The equivalence between the set of SAS states and the set of AC states (as proved by Theorem 1 of [49]) means that they can both be labeled by $\mathcal{A}_{\text{sym}}$, and both sets will satisfy the results deduced in the subsequent sections. From now on, we only use the terminology of SAS states in the symmetric sector $\mathcal{H}_1^{\vee N}$ for simplicity.

## 2.3 AS states for $2 \times m$ bipartite systems

To highlight the difference between SAS and AS states, let us first consider the case of two qubits. A direct consequence of Eq. (2) is that a two-qubit state $\rho$, with eigenspectrum $\lambda_1 \geq \lambda_2 \geq \lambda_3 \geq \lambda_4$, is AS when

$$(\lambda_1 - \lambda_3)^2 - 4\lambda_2\lambda_4 \leq 0. \tag{7}$$

In particular, a two-qubit state $\rho$ cannot be AS if it has more than one zero eigenvalue, as then Eq. (7) cannot be fulfilled. For exactly one zero eigenvalue ($\lambda_4 = 0$), the state is AS if $(\lambda_1 - \lambda_3)^2 \leq 0$, which is only possible when $\lambda_1 = \lambda_2 = \lambda_3 = 1/3$. Since symmetric two-qubit states $\rho_S$ are of rank 3 at most (they have no component on the antisymmetric state), they cannot be AS with respect to their full $SU(4)$ orbit, except for the aforementioned eigenspectrum, corresponding to the maximally mixed state in the symmetric subspace. In contrast, we will see in Sec. 3 that the picture is much richer when we restrict to the symmetric subspace, leaving room for a continuous two-dimensional set $\mathcal{A}_{\text{sym}}$.

For the general case of a $2 \times m$ bipartite state, we can use a result of Johnston [50] which states that a state $\rho$ with spectrum $\lambda_1 \geq \lambda_2 \geq \dots \geq \lambda_{2m} \geq 0$ is AS if and only if

$$\lambda_1 \leq \lambda_{2m-1} + 2\sqrt{\lambda_{2m-2}\lambda_{2m}}. \tag{8}$$

The previous condition is equivalent to the absolute PPT criterion of a state [18], as it is proved in [50]. For an $N$-qubit state $\rho$ viewed as a $2 \times 2^{N-1}$ bipartite state $\rho$, yields $\rho$ is AS if and only if $\lambda_1 \leq \lambda_{2^N-1} + 2\sqrt{\lambda_{2^N-2}\lambda_{2^N}}$. But for symmetric states, which have support only on the symmetric subspace, $\lambda_k = 0 \ \forall \ k > N+1$, which leads to the condition $\lambda_1 \leq 0$ that can never be fulfilled since $\lambda_1 > 0$ is the largest eigenvalue of $\rho$. Although symmetric multiqubit states of more than two qubits cannot be AS, they can be SAS as we show below. A trivial example is the maximally mixed state in the symmetric sector for the three-qubit system $\rho \in \mathcal{B}(\mathcal{H}_1^{\vee 3}) \subset \mathcal{B}(\mathcal{H}_1 \otimes \mathcal{H}_1^{\vee 2})$ which, seen as a $2 \times 3$ system, has a spectrum consisting of six eigenvalues, two of which are zero ($\lambda_5 = \lambda_6 = 0$). Hence, it is not AS but is evidently SAS.

## 3 Symmetric two-qubit states

### 3.1 Maximum negativity

The central question presented in the introduction can now be reformulated as follows: For a symmetric two-qubit mixed state $\rho_S$, what is the maximum entanglement that can be obtained by a global unitary transformation $U_S \in SU(3)$ that leaves the symmetric sector invariant ? The answer to this question is stated by the following theorem:

**Theorem 1** *Let $\rho_S$ be a symmetric two-qubit state with spectrum $\tau_1 \geq \tau_2 \geq \tau_3$. It holds that*

$$\max_{U_S \in SU(3)} \mathcal{N}\left(U_S \rho_S U_S^\dagger\right) = \max\left(0, \sqrt{\tau_1^2 + (\tau_2 - \tau_3)^2} - \tau_2 - \tau_3\right), \tag{9}$$

*where the maximum negativity is reached by the state $\tilde{\rho}_S = U_S \rho_S U_S^\dagger$ given up to local unitary transformations by*

$$\tilde{\rho}_S = \tau_3 |D_2^{(0)}\rangle\langle D_2^{(0)}| + \tau_1 |D_2^{(1)}\rangle\langle D_2^{(1)}| + \tau_2 |D_2^{(2)}\rangle\langle D_2^{(2)}|. \tag{10}$$

The full characterization of the set $\mathcal{A}_{\text{sym}}$ follows immediately from Theorem 1:

**Corollary 1** *Let $\rho_S$ be a symmetric two-qubit state with spectrum $\tau_1 \geq \tau_2 \geq \tau_3$. Then $\rho_S \in \mathcal{A}_{\text{sym}}$ if and only if its eigenvalue spectrum fulfills*

$$\sqrt{\tau_2} + \sqrt{\tau_3} \geq 1. \tag{11}$$

*Proof.* The symmetric two-qubit state $\rho_S$ is SAS if the right-hand side of (9) is zero which, using the normalization condition $\tau_1 + \tau_2 + \tau_3 = 1$, is equivalent to the above inequality. $\square$

In Fig. 1, we show a density plot of the maximum negativity given by Eq. (9) for all states $\rho_S \in \mathcal{B}(\mathcal{H}_1^{\vee 2})$ in terms of its two smallest eigenvalues $\tau_2$ and $\tau_3$. The solid lines correspond to the states with two coincident spectrum eigenvalues, $\tau_2 = \tau_3$ or $\tau_1 = \tau_2$, respectively. The 2-dimensional white region constitutes the set $\mathcal{A}_{\text{sym}}$ and its boundaries are given by two inequalities associated with the eigenvalues sorting, $\tau_2 \geq \tau_3$ and $2\tau_2 + \tau_3 \leq 1$ (solid lines), and Eq. (11) (black dashed line). The end (yellow) points $q_1$ and $q_2$ (of the boundary (11)) correspond to the spectra $(\tau_1, \tau_2, \tau_3) = (4/9, 4/9, 1/9)$ and $(1/2, 1/4, 1/4)$, respectively. We also remark that when $\rho_S$ has one zero eigenvalue $\tau_3 = 0$, the condition (11) cannot be met and then $\rho_S \notin \mathcal{A}_{\text{sym}}$, as can be seen in Fig. 1. The characterization of the SAS states for the symmetric two-qubit system was studied recently in [31], where they also reported the same characterization of SAS states given in our Corollary 1.

Another famous measure of entanglement for two qubits is the concurrence $C(\rho)$, defined as [51]

$$C(\rho) = \max(0, s_1 - s_2 - s_3 - s_4), \tag{12}$$

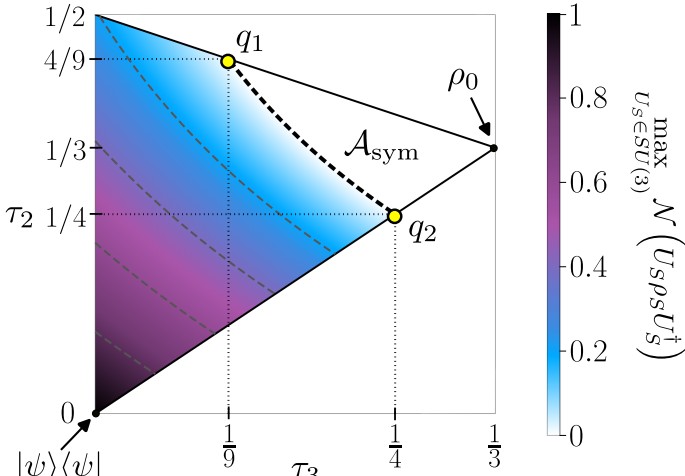

Figure 1: Density plot of the maximum negativity (9) attained in the $SU(3)$ orbit of a symmetric two-qubit state $\rho_S$ over the simplex of its eigenvalues $(\tau_3, \tau_2)$. The contour curves are shown for a maximum negativity equal to $0.8, 0.6, 0.4, 0.2$ (grey dashed lines). The set $\mathcal{A}_{\text{sym}}$ is depicted by the white region bounded on the left by the black dashed curve (which follows from Corollary 1), with end points $q_1$ and $q_2$ (yellow). The right corner of the simplex corresponds to the maximally mixed symmetric state, $\rho_0 = \mathbb{1}_3/3 \in \mathcal{B}(\mathcal{H}_1^{\vee 2})$, and the lower corner to pure states.

where $s_i$ are the singular values of the matrix $\sqrt{\rho^T} S \sqrt{\rho}$ with $S = \sigma_y \otimes \sigma_y$ and $\sigma_y$ the second Pauli matrix. Interestingly, the state (10) also maximizes the concurrence in its respective $SU(3)$ orbit, with a value equal to

$$\max_{U_S \in SU(3)} C\left(U_S \rho_S U_S^\dagger\right) = \max\left(0, \tau_1 - 2\sqrt{\tau_2 \tau_3}\right). \tag{13}$$

The proof of this result is given in Appendix A.

We end this subsection with the proof of Theorem 1.

*Proof of Theorem 1.* The partial transpose of a qubit-qubit state has at most one negative eigenvalue [52], reducing the expression of the negativity to

$$\mathcal{N}(\rho) = 2 \max\left(0, -\Lambda_{\min}\right), \tag{14}$$

with $\Lambda_{\min}$ the minimal eigenvalue of $\rho_S^{T_A}$. Moreover, $\Lambda_{\min}$ is equivalent to [15, 31]

$$\Lambda_{\min} = \min_{|\psi\rangle \in \mathcal{H}_1^{\otimes 2}} \text{Tr}\left[\rho_S(|\psi\rangle\langle\psi|)^{T_A}\right]. \tag{15}$$

The general two-qubit state $|\psi\rangle$ can be written as a linear superposition of a symmetric state and an antisymmetric state, $|\psi_S\rangle \in \mathcal{H}_1^{\vee 2}$ and $|\psi_A\rangle \in \mathcal{H}_1^{\wedge 2}$,

$$|\psi\rangle = \cos\alpha |\psi_S\rangle + e^{i\delta} \sin\alpha |\psi_A\rangle, \tag{16}$$

where $\alpha \in [0, \pi/2]$ and $\delta \in [0, 2\pi]$. On the one hand, $|\psi_S\rangle$ can be written via the Schmidt decomposition as

$$|\psi_S\rangle = \cos\beta |n_1\rangle \otimes |n_1\rangle + \sin\beta |n_2\rangle \otimes |n_2\rangle, \tag{17}$$

where $\Gamma = \{|n_j\rangle\}_{j=1}^2$ is an orthogonal basis of $\mathcal{H}_1$ and $\beta \in [0, \pi/2]$.[1] On the other hand, $|\psi_A\rangle$ can be written for any orthogonal basis of $\mathcal{H}_1$, in particular $\Gamma$, as

$$|\psi_A\rangle = \frac{1}{\sqrt{2}}\left(|n_1\rangle \otimes |n_2\rangle - |n_2\rangle \otimes |n_1\rangle\right). \tag{18}$$

---

[1] Although the Schmidt coefficients are always positive, we consider here that the coefficients can have different signs.

Hence, $|\psi\rangle\langle\psi|$ is the sum of four terms

$$|\psi\rangle\langle\psi| = \cos^2\alpha\,|\psi_S\rangle\langle\psi_S| + \sin^2\alpha\,|\psi_A\rangle\langle\psi_A| + \cos\alpha\sin\alpha\left(e^{-i\delta}|\psi_S\rangle\langle\psi_A| + e^{i\delta}|\psi_A\rangle\langle\psi_S|\right). \quad (19)$$

The condition for a state $\rho_S$ to be symmetric is that it has support only on the symmetric sector of $\mathcal{B}(\mathcal{H}_1^{\otimes 2})$, which can be written as $\rho_S = P_S\rho_S P_S$ with $P_S$ the projection operator onto $\mathcal{H}_1^{\vee 2}$. For convenience, we now introduce the symmetrized state $|n_1, n_2\rangle$ resulting from the action of $P_S$ on the product state $|n_1\rangle \otimes |n_2\rangle$,

$$P_S|n_1\rangle \otimes |n_2\rangle = \frac{1}{2}\left(|n_1\rangle \otimes |n_2\rangle + |n_2\rangle \otimes |n_1\rangle\right) \equiv \frac{|n_1, n_2\rangle}{\sqrt{2}}.$$

Replacing $\rho_S$ in Eq. (15) by $P_S\rho_S P_S$ and using the cyclic property of the trace, we get

$$\Lambda_{\min} = \min_{|\psi\rangle\in\mathcal{H}_1^{\otimes 2}} \text{Tr}\left[\rho_S P_S(|\psi\rangle\langle\psi|)^{T_A}P_S\right] = \min_X \text{Tr}\left[\rho_S X\right],$$

where the operator $X = P_S(|\psi\rangle\langle\psi|)^{T_A}P_S$ can be developed as

$$X = \cos^2\alpha\,\Sigma_1 + \sin^2\alpha\,\Sigma_2 + \cos\alpha\sin\alpha\sin\delta\,\Sigma_3, \quad (20)$$

where the $\Sigma_j$ operators are represented in the orthonormal basis $\Gamma' = \{|n_1\rangle^{\otimes 2}, |n_1, n_2\rangle, |n_2\rangle^{\otimes 2}\}$ by the matrices

$$\Sigma_1 = \begin{pmatrix} \cos^2\beta & 0 & 0 \\ 0 & \cos\beta\sin\beta & 0 \\ 0 & 0 & \sin^2\beta \end{pmatrix}, \quad \Sigma_2 = \frac{1}{2}\begin{pmatrix} 0 & 0 & -1 \\ 0 & 1 & 0 \\ -1 & 0 & 0 \end{pmatrix}, \quad (21)$$

$$\Sigma_3 = i\begin{pmatrix} 0 & \cos\beta & 0 \\ -\cos\beta & 0 & \sin\beta \\ 0 & -\sin\beta & 0 \end{pmatrix}. \quad (22)$$

The basis $\Gamma'$ can be transformed to the symmetric Dicke basis $\{|D_2^{(k)}\rangle\}_{k=0}^2$ by a diagonal $SU(2) \times SU(2)$-transformation $V = \tilde{V} \otimes \tilde{V}$ such that $\tilde{V}^\dagger|n_1\rangle = |+\rangle$ and $\tilde{V}^\dagger|n_2\rangle = |-\rangle$. Hence, $P_S(|\psi\rangle\langle\psi|)^{T_A}P_S$ for a general state $|\psi\rangle$ is parametrized by the $(\alpha, \beta, \delta)$ variables and a diagonal $SU(2) \times SU(2)$-transformation $V$

$$P_S(|\psi\rangle\langle\psi|)^{T_A}P_S = V^\dagger X V, \quad (23)$$

where the matrix $X$ written in the symmetric Dicke basis has the form (20). The smallest value of $\Lambda_{\min}$ over the $SU(3)$ orbit of $\rho_S$ is equal to

$$\min_{U\in SU(3)} \Lambda_{\min} = \min_{U,V,\alpha,\beta,\delta} \text{Tr}\left[U\rho_S U^\dagger V^\dagger X V\right]. \quad (24)$$

Without loss of generality, the $U$ and $V$ unitary transformations can be combined to $W = VU$ because the diagonal $SU(2) \times SU(2)$ transformation $V$ is in the $SU(3)$ group.[2] The minimization problem then reduces to

$$\min_{\substack{W\in SU(3) \\ \alpha,\beta,\gamma}} \text{Tr}\left[\rho_S W^\dagger X W\right]. \quad (25)$$

---

[2]A diagonal $SU(2) \times SU(2)$ operation is equivalent to a rotation times a global phase factor, which are also operations contained in $SU(3)$.

Birkhoff's theorem (Theorem 8.7.2 of [53]) establishes that the minimum over all $W \in SU(3)$ is attained when $W$ is the product of matrices diagonalizing $\rho_S$ and $X$ in the same basis, and a matrix $W(\pi)$ representing a permutation $\pi \in S_3$ where $S_3$ is the permutation group of three elements. Without loss of generality, we consider $\rho_S$ and $X$ to be represented by the diagonal matrices $\rho_d$ and $X_d$ in the Dicke basis. Thus, (25) is given by

$$\min_{\substack{\pi \in S_3 \\ \alpha, \beta, \gamma}} \mathrm{Tr}\big[\rho_d W^\dagger(\pi) X_d W(\pi)\big] = \min_{\substack{\pi \in S_3 \\ \alpha, \beta, \gamma}} \sum_{k=1}^{3} \tau_{\pi(k)} \xi_k \,,$$

where $\xi_k$ are the eigenvalues of $X$. The eigenvalues $\xi_k$ cannot generally be expressed in a compact way. However, the function to minimize in the last equation must have its derivative with respect to $\delta$ equal to zero at $W, \alpha, \beta, \delta$ where the minimum is attained, which implies that

$$\mathrm{Tr}\big[\rho_S W^\dagger \Sigma_3 W\big] \cos\alpha \sin\alpha \cos\delta = 0\,. \tag{26}$$

The latter equation is satisfied either by one of the following solutions: (A) $\delta = \pi/2, 3\pi/2$, or (B) $\alpha = 0, \pi/2$, or (C) when $\mathrm{Tr}[\rho_S W^\dagger \Sigma_3 W] = 0$. First, the eigenvalues for the solution (A) are the same for both values of $\delta$ and equal to

$$\begin{pmatrix} \xi_1 \\ \xi_2 \\ \xi_3 \end{pmatrix} = \frac{1}{2} \begin{pmatrix} 1 + \sqrt{1 - z^2} \\ z \\ 1 - \sqrt{1 - z^2} \end{pmatrix}, \tag{27}$$

with $z = -\sin^2\alpha + \cos^2\alpha \sin(2\beta)$. On the other hand, while the solution (B) keeps only $\Sigma_1$ or $\Sigma_2$ in $X$ [see Eq. (20)], the solution (C) restricts the available set of the $W$ matrices to $\mathcal{R}_\beta = \{W \in SU(3) | \mathrm{Tr}[\rho_S W^\dagger \Sigma_3 W] = 0\}$ with $\Sigma_3 = \Sigma_3(\beta)$. Then, Eq. (25) for the solution (C) is reduced and lower bounded by

$$\min_{\substack{W' \in \mathcal{R}_\beta \\ \alpha, \beta, \gamma}} \mathrm{Tr}\big[\rho_S W'^\dagger X W'\big] = \min_{\substack{W' \in \mathcal{R}_\beta \\ \alpha, \beta, \gamma}} \mathrm{Tr}\big[\rho_S W'^\dagger(\cos^2\alpha \Sigma_1 + \sin^2\alpha \Sigma_2) W'\big]$$

$$\geq \min_{\substack{W \in SU(3) \\ \alpha, \beta, \gamma}} \mathrm{Tr}\big[\rho_S W^\dagger(\cos^2\alpha \Sigma_1 + \sin^2\alpha \Sigma_2) W\big]\,. \tag{28}$$

Thus, the solution (B) and the upper bound of (C) can be studied simultaneously by omitting $\Sigma_3$ in the minimization problem, leaving $X = \cos^2\alpha \Sigma_1 + \sin^2\alpha \Sigma_2$ with eigenvalues equal to

$$\begin{pmatrix} \xi_1 \\ \xi_2 \\ \xi_3 \end{pmatrix} = \frac{1}{4} \begin{pmatrix} 1 + y_1 - \sqrt{2(1 + y_1^2 - 2y_2^2)} \\ 1 + y_1 + \sqrt{2(1 + y_1^2 - 2y_2^2)} \\ 1 - y_1 + 2y_2 \end{pmatrix}, \tag{29}$$

where $y_1 = \cos(2\alpha)$ and $y_2 = \cos^2\alpha \sin(2\beta)$. For both sets of eigenvalues (27) and (29), we must now find the critical points of (26) with respect to the variables $\alpha$ and $\beta$. We enlist in Appendix B all the critical points obtained for the cases mentioned above. By comparing the values obtained for (26) with all the possible permutations $\pi$, we deduce that the minimum $\Lambda_{\min}$ in the $SU(3)$ orbit of $\rho_S$ is reached for the solution (A) with

$$z = -\sin^2\alpha + \cos^2\alpha \sin(2\beta) = -\frac{\tau_1}{\sqrt{\tau_1^2 + (\tau_2 - \tau_3)^2}}\,, \tag{30}$$

and with $\pi$ such that

$$\Lambda_{\min} = \tau_3\,\xi_1 + \tau_1\,\xi_2 + \tau_2\,\xi_3 = \frac{1}{2}\left(\tau_2 + \tau_3 - \sqrt{\tau_1^2 + (\tau_2 - \tau_3)^2}\right). \tag{31}$$

It is this value of $\Lambda_{\min}$ which gives the expression (9) for the negativity $\mathcal{N}(\rho_S)$. In particular, for $\alpha = 0$ and $\sin(2\beta) = -\tau_1/\sqrt{\tau_1^2 + (\tau_2 - \tau_3)^2}$, the $X$ matrix is already diagonal in the symmetric Dicke basis and reads

$$X = \xi_1|D_2^{(0)}\rangle\langle D_2^{(0)}| + \xi_2|D_2^{(1)}\rangle\langle D_2^{(1)}| + \xi_3|D_2^{(2)}\rangle\langle D_2^{(2)}|. \tag{32}$$

In order to attain (31), $\rho_S$ must then be equal to (10), up to a local unitary transformation. $\square$

Let us remark that the minimization in (15) is performed over all states $|\psi\rangle \in \mathcal{H}_1^{\otimes 2}$, and we found that the states $|\psi\rangle$ that minimize $\Lambda_{\min}$ over the $SU(3)$ orbit of $\rho_S$ are of the form (16), with $\delta = \pi/2, 3\pi/2$ and $(\alpha, \beta)$ such that Eq. (30) is satisfied. This implies that there exists a 1-dimensional set of states $|\psi\rangle \in \mathcal{H}_1^{\otimes 2}$ that minimize $\Lambda_{\min}$ of $\rho_S$. In particular, for $\alpha = 0$, the state $|\psi\rangle$ belongs to the symmetric sector $\mathcal{H}_1^{\vee 2}$.

## 3.2 Extension of the set of SAS states

In this subsection, we derive, from the previous results, the radii of the maximal ball contained in $\mathcal{A}_{\text{sym}}$ and the minimal ball that includes $\mathcal{A}_{\text{sym}}$, both centred on the maximally mixed state in the symmetric subspace $\rho_0 = (N+1)^{-1}\mathbb{1}_{N+1}$ with $\mathbb{1}_{N+1}$ the identity matrix of size $N+1$. We denote by $r$ the Hilbert-Schmidt distance between a state $\rho_S$ and $\rho_0$

$$r \equiv \left\|\rho_S - \rho_0\right\|_{\text{HS}} = \sqrt{\text{Tr}\left[\rho_S^2\right] - \frac{1}{N+1}}. \tag{33}$$

The range for $r$ is $\left[0, \sqrt{\frac{N}{N+1}}\right]$, where the upper and lower bounds are reached when $\rho_S$ is equal to $\rho_0$ or a pure state, respectively. As in the non-symmetric case, there are balls centred on $\rho_0$ containing only SAS states. Consequently, there exists a maximum radius $r_{\text{SAS}}$ such that any ball centred on $\rho_0$ with radius $r \le r_{\text{SAS}}$ contains only SAS states. A lower bound $r_{\text{SAS}}^{\text{LB}}$ for $r_{\text{SAS}}$ has been determined in Ref. [32] (in the context of the absolute classicality of spin-$s$ states with $s = N/2$)

$$r_{\text{SAS}}^{\text{LB}} = \frac{1}{\sqrt{(N+1)\left[2(2N+1)\binom{2N}{N} - (N+2)\right]}}. \tag{34}$$

To determine the exact value of $r_{\text{SAS}}$, we first calculate the distance $r$ for the states (10) that maximize the negativity in each $SU(3)$ orbit. Using the normalization condition $\tau_1 + \tau_2 + \tau_3 = 1$, we obtain

$$r^2 = \frac{2}{3} + 2\left(\tau_2^2 + \tau_3^2 + \tau_2\tau_3 - \tau_2 - \tau_3\right). \tag{35}$$

The last equation allows us to express $\tau_2$ in terms of $r$ and $\tau_3$, and thus the maximum negativity over each $SU(3)$ orbit of $\rho_S$, Eq. (9), as a function of $\tau_3$ and $r$. In Fig. 2, we show a density plot of this maximum negativity, where the variables $\tau_3$ and $r$ are subject to the constraints $\sqrt{2/3}(1 - 3\tau_3) \ge r \ge (1 - 3\tau_3)/\sqrt{6}$ depicted by straight lines. The black dashed curve delimiting $\mathcal{A}_{\text{sym}}$ (white region) corresponds to the inequality

$$r \le \sqrt{\frac{2}{3}}\left[1 + 3\left(\tau_3 - \sqrt{\tau_3}\right)\right], \quad \tau_3 \in \left[\frac{1}{9}, \frac{1}{3}\right]. \tag{36}$$

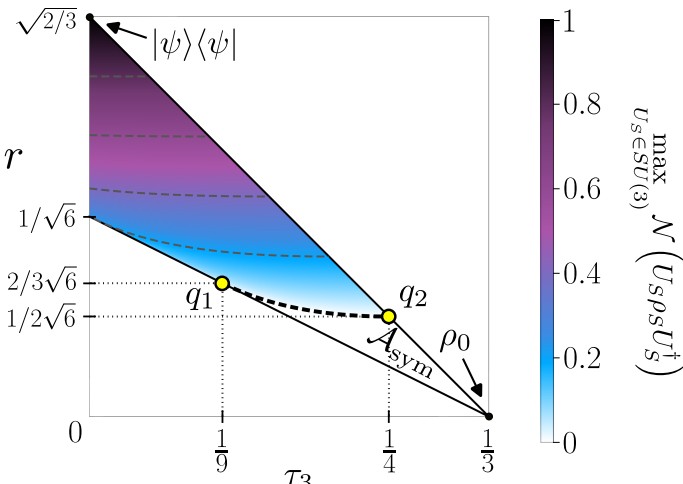

Figure 2: Density plot of the maximum negativity (9) over the simplex of symmetric two-qubit states $\rho_S$ parametrized with $(\tau_3, r)$. The $\mathcal{A}_{\text{sym}}$ boundary is now given by Eq. (36).

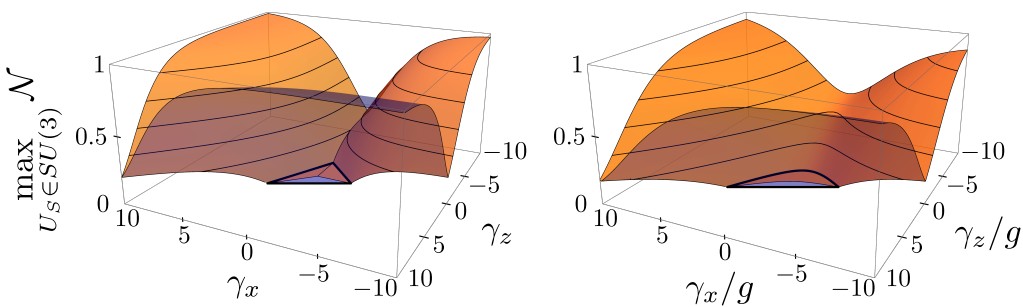

Figure 3: Maximum negativity in the SU(3) orbit of a spin-1 thermal state with Hamiltonian (38) with (left) $k_B T/\hbar = 2$ (in the same units of frequency as $\gamma_x$ and $\gamma_z$) for $g = 0$, and (right) $k_B T/\hbar g = 3$ for $g \neq 0$. The black thick lines show the bounds of the $(\gamma_x, \gamma_z)$ values where the corresponding thermal states $\rho_S$ are SAS.

The end points of the dashed curve corresponding to the (yellow) points $q_1$ and $q_2$ shown in Fig. 1, have coordinates $(\tau_3, r)$

$$\left(\frac{1}{9}, \frac{2}{3\sqrt{6}}\right), \quad \text{and} \quad \left(\frac{1}{4}, \frac{1}{2\sqrt{6}}\right), \tag{37}$$

respectively. As a result, all states with $r \leqslant 1/(2\sqrt{6})$ are necessarily SAS, from which we deduce that $r_{\text{SAS}} = 1/(2\sqrt{6})$. This value is strictly larger than that provided by Eq. (34), equal to $1/(2\sqrt{42})$ for $s = N/2 = 1$. Moreover, we can easily obtain the radius $R_{\text{SAS}}$ of the smallest ball containing $\mathcal{A}_{\text{sym}}$ by observing in Fig. 2 that the SAS state furthest from $\rho_0$ has a spectrum associated with the point $q_1$. Hence, we can conclude that $R_{\text{SAS}} = 2/(3\sqrt{6})$ and that any state $\rho_S$ at a distance from $\rho_0$ larger than $R_{\text{SAS}}$ cannot be SAS.

## 3.3 Spin-1 system at finite temperature

We now apply our results to determine the maximum achievable entanglement from the sole application of a unitary transformation on a thermal state. We consider a spin-1 system at

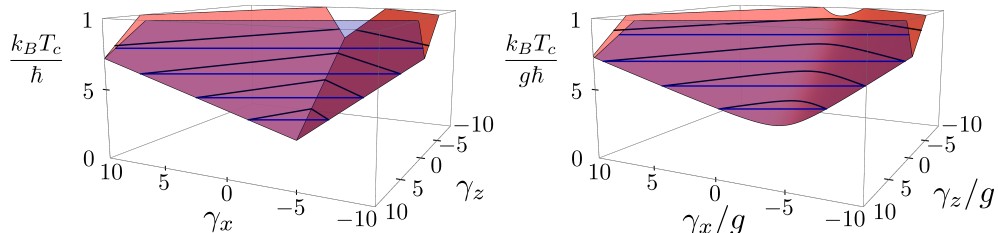

Figure 4: Critical temperature $T_c$ above which the thermal state of the Hamiltonian (38) becomes SAS. We plot the cases for (left) $g = 0$ (with $k_B T_c/\hbar$, $\gamma_x$ and $\gamma_z$ in the same units of frequency) and (right) $g \neq 0$.

temperature $T$ with a Hamiltonian of the form

$$H/\hbar = g\tilde{S}_z + \gamma_x \tilde{S}_x^2 + \gamma_z \tilde{S}_z^2 , \tag{38}$$

where $\tilde{S}_n \equiv S_n/\hbar$ are the (dimensionless) angular momentum operators along the $n = x, y, z$ axes, and $g$, $\gamma_x$ and $\gamma_z$ are coupling strengths (in the same units of frequency). Without loss of generality, we can consider $g \geq 0$. The Hamiltonian (38) appears in several bosonic systems, such as Bose-Einstein condensates [54] or the Lipkin-Meshkov-Glick model [37,38]. The corresponding thermal state $\rho_S$ of the system has normalized eigenspectrum equal to

$$\tau_k = \frac{e^{-\beta \epsilon_{4-k}}}{Z} , \quad \text{with} \quad Z = \text{Tr}\left(e^{-\beta H}\right) , \tag{39}$$

with $\epsilon_k$ the eigenvalues of $H$ sorted in nonincreasing order $\epsilon_1 \geq \epsilon_2 \geq \epsilon_3$, and $\beta = 1/k_B T$ where $k_B$ is the Boltzmann constant. For $s = 1$, the eigenvalues $\epsilon_k$ are given by

$$\left\{ \frac{\hbar}{2}\left(\gamma_x + 2\gamma_z - \sqrt{4g^2 + \gamma_x^2}\right), \hbar\gamma_x, \frac{\hbar}{2}\left(\gamma_x + 2\gamma_z + \sqrt{4g^2 + \gamma_x^2}\right) \right\} , \tag{40}$$

where the order of the eigenvalues depends on the values of the coupling strengths. We plot in Fig. 3 the maximum negativity (9) in the unitary orbit of the thermal state at finite temperature as a function of the coupling strengths $\gamma_x$ and $\gamma_z$ for $g = 0$ (left) and $g \neq 0$ (right). We observe a region in the $(\gamma_x, \gamma_z)$ parameter space where the corresponding thermal states $\rho_S$ are SAS. Its boundaries can be obtained analytically by substituting the eigenspectrum of the thermal state in the condition (11), yielding

$$2\epsilon_3 - \epsilon_1 - \epsilon_2 + 2k_B T \ln 2 \geq 0 . \tag{41}$$

For $g = 0$, the eigenenergies are $(\gamma_x, \gamma_z, \gamma_x + \gamma_z)$ and the SAS region is a triangle with boundaries given by

$$\begin{cases} \hbar(\gamma_x + \gamma_z) + 2k_B T \ln 2 \geq 0, & \text{for} \quad \gamma_x, \gamma_z \leq 0, \\ \hbar(\gamma_z - 2\gamma_x) + 2k_B T \ln 2 \geq 0, & \text{for} \quad 0 \leq \gamma_z \leq \gamma_x, \text{ or } \gamma_z \leq 0 \leq \gamma_x, \\ \hbar(\gamma_x - 2\gamma_z) + 2k_B T \ln 2 \geq 0, & \text{for} \quad 0 \leq \gamma_x \leq \gamma_z, \text{ or } \gamma_x \leq 0 \leq \gamma_z. \end{cases} \tag{42}$$

On the other hand, for $g \neq 0$, the region is bounded by linear and nonlinear conditions of the $(\gamma_x, \gamma_z)$ variables

$$\begin{cases} \hbar(\gamma_x - 2\gamma_z) + 2k_B T \ln 2 \geq 0, & \text{for} \quad g^2 \leq \gamma_z(\gamma_z - \gamma_x), \text{ and } \gamma_x < 2\gamma_z, \\ \hbar\left(2\gamma_z - \gamma_x - 3\sqrt{4g^2 + \gamma_x^2}\right) + 4k_B T \ln 2 \leq 0, & \text{for} \quad g^2 \geq \gamma_z(\gamma_z - \gamma_x), \text{ or } 2\gamma_z < \gamma_x. \end{cases} \tag{43}$$

For any set of values of the coupling strengths $(g, \gamma_x, \gamma_z)$, there exists a critical temperature $T_c$ above which the thermal state $\rho_S$ is SAS. We plot in Fig. 4 the critical temperature $T_c$ as a function of the coupling strengths $(\gamma_x, \gamma_z)$, calculated by setting to zero the inequalities (42) and (43), respectively. We can observe that the contour curves of Fig. 4 are the boundaries of the SAS states appearing in Fig. 3 for the corresponding temperature.

# 4 Symmetric three-qubit states

## 4.1 Numerical results

The determination of the maximally entangled state in the $SU(4)$ orbit of a symmetric three-qubit state $\rho_S \in \mathcal{B}(\mathcal{H}_1^{\vee 3})$ can again be formulated as an optimization problem with the negativity as objective function because the PPT criterion is both a necessary and sufficient condition for entanglement in the qubit-qutrit system for which $\mathcal{H}_1^{\vee 3}$ is a subspace, see e.g. [55]. However, the optimisation is much more difficult in this case, as it must a priori be performed on the fifteen parameters of global unitary transformations, and remains an open problem at this stage. One main difference with respect to the two-qubit case is that $\rho^{T_A}$ can have one or two negative eigenvalues [52]. We have performed intensive numerical calculations on the basis of which we found a 3-parameter family of global unitary transformations $\tilde{U}_S$ which we conjecture allows the maximum negativity to be reached in the full SU(4) orbit of any state $\rho_S \in \mathcal{B}(\mathcal{H}_1^{\vee 3})$. The parametric unitary $\tilde{U}_S = \tilde{U}_S(\alpha_1, \alpha_2, \alpha_3)$ with $\alpha_j \in \mathbb{R}$ is real and has the following form in the Dicke-basis $\{|D_3^{(k)}\rangle : k = 0, \ldots, 3\}$,

$$\tilde{U}_S = \begin{pmatrix} 0 & -n_{1x}n_{2y} & n_{1y}n_{3x} - n_{1x}n_{2x}n_{3y} & -n_{1x}n_{2x}n_{3x} - n_{1y}n_{3y} \\ n_{1x} & n_{1y}n_{2x} & -n_{1y}n_{2y}n_{3y} & -n_{1y}n_{2y}n_{3x} \\ 0 & n_{1y}n_{2y} & n_{1x}n_{3x} + n_{1y}n_{2x}n_{3y} & n_{1y}n_{2x}n_{3x} - n_{1x}n_{3y} \\ n_{1y} & -n_{1x}n_{2x} & n_{1x}n_{2y}n_{3y} & n_{1x}n_{2y}n_{3x} \end{pmatrix}, \qquad (44)$$

in terms of the components of the three real unit vectors

$$\mathbf{n}_j = (n_{jx}, n_{jy}) = (\cos\alpha_j, \sin\alpha_j), \quad \text{for } j = 1, 2, 3. \qquad (45)$$

Our findings lead us to the following conjecture:

**Conjecture 1** *Let $\rho_S \in \mathcal{B}(\mathcal{H}_1^{\vee 3})$ with eigenspectrum $\tau_1 \geqslant \tau_2 \geqslant \tau_3 \geqslant \tau_4$. It holds that*

$$\max_{U_S \in SU(4)} \mathcal{N}\left(U_S \rho_S U_S^\dagger\right) = \max_{\pi, \alpha_1, \alpha_2, \alpha_3} \mathcal{N}\left(\tilde{U}_S \rho_S^\pi \tilde{U}_S^\dagger\right), \qquad (46)$$

*where $\tilde{U}_S$ is given by Eq. (44), $\rho_S^\pi$ is the diagonal state in the Dicke basis given by*

$$\rho_S^\pi = \sum_{k=0}^{3} \tau_{\pi(k+1)} |D_3^{(k)}\rangle\langle D_3^{(k)}|, \qquad (47)$$

*and $\pi$ is a permutation of the eigenvalues.*

We have successfully tested the validity of the Conjecture 1 on 24,000 states by sampling their unitary orbits with 2 million randomly chosen global unitary operations and comparing the maximum obtained for the negativity with Eq. (46). We plot in Fig. 5 the maximum negativity calculated according to the Conjecture 1 in the eigenvalues simplex $(\tau_2, \tau_3, \tau_4)$, sorted in non-increasing order, for the values of $\tau_4 = 0, 1/10, 3/20, 7/38$, respectively. The cloud of pink points shows the spectra for which Eq. (46) gives zero and for which a random sampling

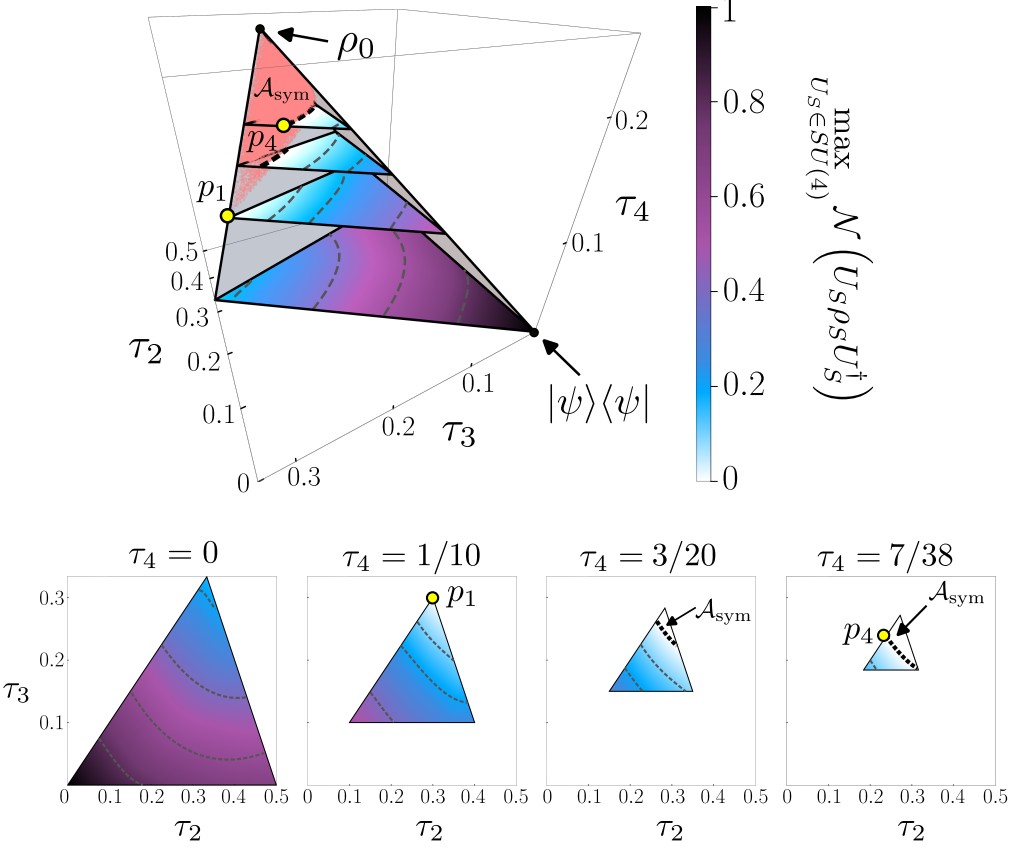

Figure 5: Density plot of the maximum negativity (computed using Conjecture 1) of symmetric three-qubit states over the simplex of eigenvalues $(\tau_2, \tau_3, \tau_4)$ for $\tau_4 = 0, 1/10, 3/20, 7/38$. The grey dashed lines are contour curves where the maximum negativity is equal to 0.8, 0.6, 0.4, 0.2, 0.1, respectively. The black dashed lines correspond to the SAS boundaries calculated by Conjecture (1), which are tangent to the set of SAS states (pink points, see text). The points $p_1$ and $p_4$, corresponding to the eigenvalues $(\tau_1, \tau_2, \tau_3, \tau_4) = (3, 3, 3, 1)/10$ and $(13, 9, 9, 7)/38$, define bounds for the radii of the balls that contains either the whole set $\mathcal{A}_{\text{sym}}$, or only SAS states.

of more than 2 million states along the global unitary orbit yielded only separable states. We can observe in Fig. 5 that the boundaries of $\mathcal{A}_{\text{sym}}$ defined by Conjecture 1 are tangent to the set of pink points. As we mention a posteriori, the points $p_1$ and $p_4$, associated to the eigenspectra $(\tau_1, \tau_2, \tau_3, \tau_4) = (3, 3, 3, 1)/10$ and $(13, 9, 9, 7)/38$, define bounds for the radii of the balls that contains either the whole set $\mathcal{A}_{\text{sym}}$ or only SAS states, respectively.

The 3-parameter family is motivated by other three particular states which reach the maximum negativity, observed numerically, over certain regions of the faces of the simplex (see Fig. 6):

*i*) For $(\alpha_1, \alpha_2, \alpha_3) = (0, 0, 0)$, the maximum negativity is achieved for the state

$$\rho_S^{(i)} = \tau_4 |D_3^{(0)}\rangle\langle D_3^{(0)}| + \tau_1 |D_3^{(1)}\rangle\langle D_3^{(1)}| + \tau_3 |D_3^{(2)}\rangle\langle D_3^{(2)}| + \tau_2 |D_3^{(3)}\rangle\langle D_3^{(3)}|, \tag{48}$$

yielding

$$\mathcal{N}\left(\rho_S^{(i)}\right) = \frac{1}{3} \max\left[0, \sqrt{8\tau_1^2 + (2\tau_3 - 3\tau_4)^2} - 2\tau_3 - 3\tau_4\right]. \tag{49}$$

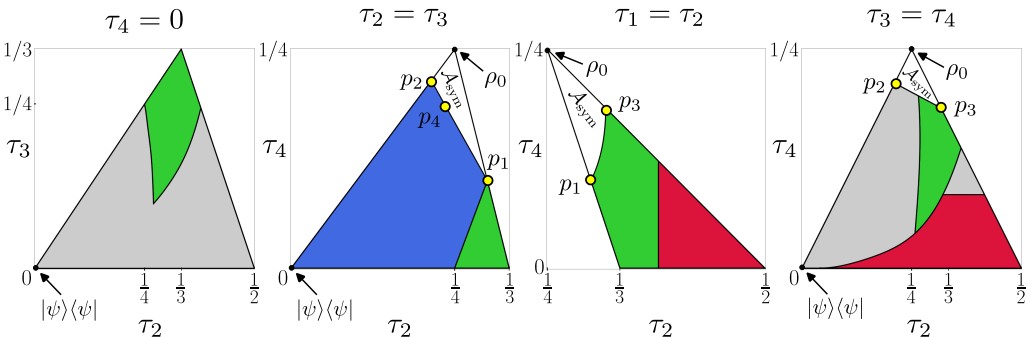

Figure 6: Regions where the states (48), (50) and (51), or a generic state defined in Conjecture 1, achieve the maximum entanglement, colored in green, blue, red or grey, respectively. We plot the faces of the simplex shown in Fig. 5, each one given by a condition in the eigenspectrum of $\rho_S$. The white regions correspond to the set $\mathcal{A}_{\text{sym}}$. The $\mathcal{A}_{\text{sym}}$ boundaries in the edges of the simplex that connect with the maximally mixed state are given by the points $p_1$, $p_2$ and $p_3$, with associated eigenspectra $(3, 3, 3, 1)/10$, $(5, 3, 3, 3)/14$ and $(3-\sqrt{3}, 3-\sqrt{3}, \sqrt{3}-1, \sqrt{3}-1)/4$, respectively. We also plot the point $p_4$ associated to the spectrum $(13, 9, 9, 7)/38$ that gives the optimal upper bound of $R_{SAS}$ (53). On the other hand, the optimal upper bound of $r_{SAS}$ is given by $p_1$ (55).

$\rho_S^{(i)}$ achieves the maximum negativity for the eigenspectra associated to the green regions shown in Fig. 6, where we plot the faces of the simplex of Fig. 5. In particular, we have observed that this basis defines the SAS boundaries in the edges of the simplex $\tau_1 = \tau_2 = \tau_3$ and $\tau_1 = \tau_2 \vee \tau_3 = \tau_4$, which are the points $p_1$ and $p_3$ with associated eigenspectra $(3, 3, 3, 1)/10$ and $(3-\sqrt{3}, 3-\sqrt{3}, \sqrt{3}-1, \sqrt{3}-1)/4$.

*ii*) For $(\alpha_1, \alpha_2, \alpha_3) = (\pi/6, 0, 0)$, the state is equal, up to a global rotation by $\pi/2$ along the $y$ axis, to

$$\rho_S^{(ii)} = V\rho_S^{(i)}V^\dagger, \quad \text{with} \quad V = \frac{1}{\sqrt{2}}\begin{pmatrix} 1 & 1 & 0 & 0 \\ 0 & 0 & 1 & 1 \\ 0 & 0 & -1 & 1 \\ -1 & 1 & 0 & 0 \end{pmatrix}. \tag{50}$$

This state can have two negative eigenvalues coming from the roots of an irreducible polynomial of degree three. We have obtained numerically that $\rho_S^{(ii)}$ achieves the maximum negativity over the blue regions in Fig. (6). We proved that this state achieves the maximum negativity for the eigenspectra $\tau_2 = \tau_3 = \tau_4$ in Appendix C, where we also conclude that the point $p_2$ in the SAS boundary has spectrum equal to $(5, 3, 3, 3)/14$. The blue region also contains the points $p_1$ and $p_4$ that, as we mention below, give tight bounds for $R_{\text{SAS}}$ and $r_{\text{SAS}}$.

*iii*) Lastly, the state defined by the unitary transformation (44) with $(\alpha_1, \alpha_2, \alpha_3) = (0, \alpha_2, \pi)$ achieves the maximum negativity in the red region of Fig. 6. The corresponding state is

$$\rho_S^{(iii)} = V\rho_S^{(i)}V^\dagger, \quad \text{with} \quad V = \begin{pmatrix} -\sin\alpha_2 & \cos\alpha_2 & 0 & 0 \\ 0 & 0 & 1 & 0 \\ 0 & 0 & 0 & 1 \\ \cos\alpha_2 & \sin\alpha_2 & 0 & 0 \end{pmatrix}, \tag{51}$$

which, for $\alpha_2 = 0$ is equivalent, up to a permutation of the eigenvalues, to the $\rho_S^{(i)}$ state.

We plot in Fig. 6 the regions where each of the states listed above achieves the maximum negativity with a difference at most of $10^{-6}$ with respect to the full family of states (44). The

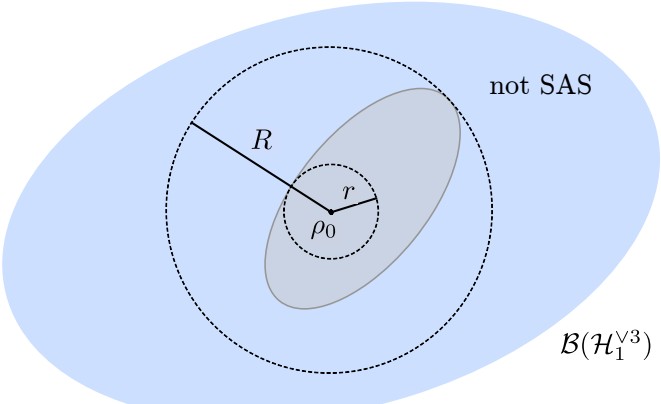

Figure 7: Sketch of the set of states $\rho_S \in \mathcal{B}(\mathcal{H}_1^{\vee 3})$ which satisfy Obs. 1, represented by the outer (blue) region. While the outer region is composed only of non-SAS states, the set $\mathcal{A}_{\text{sym}}$ is contained in the inner (grey) region. The balls of radii $R$ and $r$ define the upper bounds of $R_{\text{SAS}}$ and $r_{\text{SAS}}$, respectively.

optimal state in the grey regions are given by a generic state specified in Conjecture 1, and the white region corresponds to set $\mathcal{A}_{\text{sym}}$. It has been observed numerically that the 3-parameter family (44) also get the maximum negativity for states with eigenspectrum corresponding to points inside the simplex.

The three states mentioned above give analytical witnesses for symmetric non-absolute separability. In particular, we only need to consider the conditions given by the states $\rho_S^{(i)}$ and $\rho_S^{(ii)}$ because $\rho_S^{(iii)}$ gives a weaker condition.

**Observation 1** *A symmetric three-qubit state $\rho_S$ cannot be SAS if its eigenspectrum $\tau_1 \geqslant \tau_2 \geqslant \tau_3 \geqslant \tau_4$ satisfies*

$$\tau_1 > \sqrt{3\,\tau_3\tau_4} \quad \wedge \quad (3\tau_1 - 2\tau_2)^2\tau_3 + 3(\tau_2^2 - \tau_3^2)\tau_4 > 9\tau_3\tau_4^2. \tag{52}$$

The first expression is deduced from Eq. (49), and the second from the application of Descartes' rule of sign to the characteristic polynomial for the partial transpose of the state (50). The bounds are not tight, and a counterexample is given by a state with spectrum (0.346, 0.254, 0.2, 0.2), which does not satisfy any of the conditions listed in Obs. 1 and which has a maximum negativity $\mathcal{N} \approx 3.76 \times 10^{-4}$ for the values $(\alpha_1, \alpha_2, \alpha_3) = (2.817, \pi, 1.588)$ according to Conjecture 1. It is interesting to note that the non-SAS states escaping Observation 1 all have a negativity of order $10^{-4}$ at most, which proves the effectiveness of Eq. (52) as a witness of symmetric non-absolute separability.

## 4.2 Extension of the set of SAS states

The three-dimensional simplex associated with the spectrum eigenvalues $(\tau_2, \tau_3, \tau_4)$ is divided into two regions by the conditions of Obs. 1 as shown in Fig. 7. The outer (blue) region is composed only of non-SAS states, while the grey (inner) region contains the set $\mathcal{A}_{\text{sym}}$ and other non-SAS states. Hence, an upper bound of $R_{\text{SAS}}$ ($r_{\text{SAS}}$) is provided by the maximum (minimum) achievable distance $r$ between $\rho_0$ and states lying on the boundaries specified by Obs. 1. Following the same reasoning as in Subsec. 3.2, the minimal distance is obtained from the second condition of Obs. 1 for

$$(\tau_1, \tau_2, \tau_3, \tau_4) = \left(\frac{13}{38}, \frac{9}{38}, \frac{9}{38}, \frac{7}{38}\right), \quad \text{with} \quad r = \frac{1}{2\sqrt{19}}. \tag{53}$$

Then, $r_{\text{SAS}}$ is bounded as follows

$$\frac{1}{10\sqrt{11}} \leqslant r_{\text{SAS}} \leqslant \frac{1}{2\sqrt{19}}, \tag{54}$$

where the lower bound comes from Eq. (34). The maximum distance $R$, and therefore an upper bound of $R_{\text{SAS}}$, can be obtained by either conditions of Obs. 1 for

$$\left(\tau_1, \tau_2, \tau_3, \tau_4\right) = \left(\frac{3}{10}, \frac{3}{10}, \frac{3}{10}, \frac{1}{10}\right), \quad \text{with} \quad R_{\text{SAS}} \leqslant R = \frac{\sqrt{3}}{10}. \tag{55}$$

The points $p_1$ and $p_4$ corresponding to the eigenspectra (53) and (55) are highlighted in yellow in Figs. 5 and 6. Using a numerical procedure similar to that described in the previous subsection, the values found numerically for $R_{\text{SAS}}$ and $r_{\text{SAS}}$ are very close to the upper bounds mentioned above. We therefore conjecture that the above upper bounds for $r_{\text{SAS}}$ and $R_{\text{SAS}}$ are tight.

## 5 Conclusions

We have studied the maximum achievable negativity in global unitary orbits of generic symmetric states for two and three qubit systems. For the two-qubit case, our main result is the determination of the optimal state within its $SU(3)$ orbit, i.e. the one that reaches the maximum negativity (Theorem 1). We also proved the optimality of the state (10) for the concurrence, another common measure of two-qubit entanglement. A direct consequence of Theorem 1 is the complete characterization of the set of SAS states based on their spectrum as given by Corollary 1, and the minimal (maximal) radius of a ball containing the whole set of (only) SAS states. We have applied our results to a spin-1 system with a spin-squeezing Hamiltonian by studying the maximum entanglement that can be produced from a thermal state by a global unitary operation, and have examined its temperature dependence. In particular, we have obtained the expression for a critical temperature above which the thermal state becomes SAS. For the symmetric three-qubit system, our results, both numerical and analytical, have shown the complexity of the problem. The main difference with the two-qubit case is that the maximum negativity is no longer achieved by a state that retains the same form for all possible spectra. In particular, we have found a three-parameter family of states which we conjecture achieves maximum negativity (Conjecture 1). The family includes, among others, the two states (48) and (50) which are the basis for Observation 1, a necessary condition for being SAS. In addition, it gives upper bounds for the radii of the balls containing the set $\mathcal{A}_{\text{sym}}$ or only SAS states, respectively, for which numerical results indicate that they are tight.

## Acknowledgements

The authors thank K. Życzkowski for his correspondence.

**Author contributions** ESE and JM contributed equally to this work.

**Funding information** ESE acknowledges support from the postdoctoral fellowships of DGAPA, UNAM and the IPD-STEMA program of the University of Liège.

# A  Proof for the maximal concurrence (13)

We follow a similar proof as in Ref. [15]. The concurrence of a two-qubit state $\rho$ is given by

$$C(\rho) = \max(0, s_1 - s_2 - s_3 - s_4), \tag{A.1}$$

where $s_i$ are the singular values of the matrix $\sqrt{\rho^T} S \sqrt{\rho}$ with $S = \sigma_y \otimes \sigma_y$. Since we consider a symmetric mixed state $\rho_S$, $s_4 = 0$ and the $S$ matrix in the previous expression can be projected into the symmetric sector as

$$S_S \equiv P_S S P_S = \begin{pmatrix} 0 & 0 & -1 \\ 0 & 1 & 0 \\ -1 & 0 & 0 \end{pmatrix}, \tag{A.2}$$

where we keep only the components in the symmetric sector by working in the Dicke basis. Using the decomposition $\sqrt{\rho_S} = \Phi \Lambda^{1/2}$, where $\Phi \in U(3)$ and $\Lambda^{1/2}$ is a diagonal matrix, the maximum concurrence in the $SU(3)$ orbit of $\rho_S$ is equal to

$$\max_{U_S \in SU(3)} C\left(U_S \rho_S U_S^\dagger\right) = \max_{U_S \in SU(3)} (0, s_1 - s_2 - s_3), \tag{A.3}$$

with $s_i$ the singular values of

$$\Lambda^{1/2} \Phi^T U_S^T S_S U_S \Phi \Lambda^{1/2} = \Lambda^{1/2} U' \Lambda^{1/2}, \tag{A.4}$$

where $U' \in U(3)$. Since any unitary matrix $V$ cannot necessarily be decomposed as $U'$ is in the above equation, we have

$$\max_{U_S \in SU(3)} C(U_S \rho_S U_S^\dagger) \leq \max_{V \in U(3)} (0, s_1' - s_2' - s_3'), \tag{A.5}$$

where $s_i'$ are the singular values of $\Lambda^{1/2} V \Lambda^{1/2}$. Now, we can find an upper bound on the r.h.s. of (A.5) using the following lemmas from Refs. [56] and [57], respectively:

**Lemma 1** *Let $A \in M_{n,r}(\mathbb{C})$, $B \in M_{r,m}(\mathbb{C})$. Then,*

$$\sum_{i=1}^{k} \sigma_i(AB) \leq \sum_{i=1}^{k} \sigma_i(A) \sigma_i(B), \tag{A.6}$$

*for $k = 1, \ldots, q = \min\{n, r, m\}$.*

**Lemma 2** *Let $A \in M_n(\mathbb{C})$, $B \in M_{n,m}(\mathbb{C})$ and $i \leq i_1 < \cdots < i_k \leq n$. Then*

$$\sum_{i=t}^{k} \sigma_{i_t}(AB) \geq \sum_{t=1}^{k} \sigma_{i_t}(A) \sigma_{n-t+1}(B). \tag{A.7}$$

In our case $n = 3$, and we put $k = 1$ in Lemma 1 and $k = 2$, $i_1 = 2$, $i_3 = 3$ in Lemma 2. Subtracting the resulting inequalities then gives

$$\sigma_1(AB) - [\sigma_2(AB) + \sigma_3(AB)] \leq \sigma_1(A) \sigma_1(B) - \sigma_2(A) \sigma_2(B) - \sigma_3(A) \sigma_3(B). \tag{A.8}$$

We now set $A = \Lambda^{1/2}$ and $B = V \Lambda^{1/2}$. Both matrices have the same singular values given by the eigenvalues of $\Lambda^{1/2}$, $\sigma_i(A) = \sigma_i(B) = \sqrt{\tau_i}$. Equation (A.5) then gives

$$\max_{U_S \in SU(3)} C\left(U_S \rho_S U_S^\dagger\right) \leq \max\left(0, \tau_1 - 2\sqrt{\tau_2 \tau_3}\right). \tag{A.9}$$

Finally, we prove by direct calculation that the state (10) reaches the upper bound of the concurrence, where the singular values $s_i$ can also be calculated by the square roots of the eigenvalues of the matrix

$$\rho_S S_S \rho_S S_S = \begin{pmatrix} \tau_2\tau_3 & 0 & 0 \\ 0 & \tau_1^2 & 0 \\ 0 & 0 & \tau_2\tau_3 \end{pmatrix}, \tag{A.10}$$

so that $s_1 = \tau_1$, $s_2 = s_3 = \sqrt{\tau_2\tau_3}$, which satisfies our statement. $\square$

## B  Critical points of $\Lambda$

**Case 1:**  $\delta = \pi/2, 3\pi/2$. The $X$ matrix has the same eigenvalues for both values of $\delta$, given by Eq. (27). Setting $t_j = \tau_{\pi(j)}$, the critical points of (26) and its $\Lambda$-value are the following:

i) $t_2 \geq t_3$:

$$z = -\frac{t_1}{\sqrt{t_1^2 + (t_2 - t_3)^2}}, \qquad \Lambda = \frac{1}{2}\left(t_2 + t_3 - \sqrt{t_1^2 + (t_2 - t_3)^2}\right). \tag{B.1}$$

ii) $t_2 \leq t_3$:

$$z = \frac{t_1}{\sqrt{t_1^2 + (t_2 - t_3)^2}}, \qquad \Lambda = \frac{1}{2}\left(t_2 + t_3 + \sqrt{t_1^2 + (t_2 - t_3)^2}\right). \tag{B.2}$$

iii) $\alpha = \pi/2$:

$$\Lambda = \frac{t_2 + t_3 - t_1}{2}. \tag{B.3}$$

iv) $\alpha = 0$ and $\beta = \pi/4$:

$$\Lambda = 1/2. \tag{B.4}$$

**Case 2:**  $X$ without $\Sigma_3$. The eigenvalues of $X$ are in this case given by (29). In addition to identical solutions to the previous case, other solutions appear which we list below:

i) $t_1 \geq t_2$:

$$y_1 = \frac{t_1 + t_2 - t_3}{\sqrt{1 - 8t_1 t_2}}, \quad y_2 = -\frac{t_3}{\sqrt{1 - 8t_1 t_2}}, \quad \Lambda = \frac{1}{4}\left(1 - \sqrt{1 - 8t_1 t_2}\right). \tag{B.5}$$

ii) $t_1 \leq t_2$:

$$y_1 = \frac{t_3 - t_1 - t_2}{\sqrt{1 - 8t_1 t_2}}, \quad y_2 = \frac{t_3}{\sqrt{1 - 8t_1 t_2}}, \quad \Lambda = \frac{1}{4}\left(1 + \sqrt{1 - 8t_1 t_2}\right). \tag{B.6}$$

## C  Maximum entanglement of symmetric three-qubit states with spectrum $\tau_2 = \tau_3 = \tau_4$

In this appendix, we calculate the maximum negativity in the $SU(4)$ orbit of the states $\rho_S \in \mathcal{B}(\mathcal{H}_1^{\vee 3})$ with a triple-degenerate spectrum $\tau_1 \geqslant \tau_2 = \tau_3 = \tau_4 = x$, associated to one of the edges of the simplex in Fig. 5. The states $\rho_S$ can be written as

$$\rho_S = x \sum_{k=1}^{3} |\psi_k\rangle\langle\psi_k| + (1-3x)|\psi_4\rangle\langle\psi_4| = x\,\mathbb{1}_4 + (1-4x)|\psi_4\rangle\langle\psi_4|, \tag{C.1}$$

where $x \in [0, 1/4]$. Hence, the calculation of the maximum negativity in the $SU(4)$ orbit of $\rho_S$ is reduced to

$$\max_{U_S \in SU(4)} \mathcal{N}\left(U_S \rho_S U_S^{\dagger}\right) = \max_{|\psi\rangle \in \mathcal{H}_1^{\vee 3}} \mathcal{N}\left(x\,\mathbb{1}_4 + (1-4x)|\psi\rangle\langle\psi|\right). \tag{C.2}$$

The characteristic polynomial of a linear combination of the matrices $\mathbb{1}_4^{T_A}$ and $(|\psi\rangle\langle\psi|)^{T_A}$ is in general irreducible. Let us instead calculate the eigenvalues of each matrix individually. The eigenvalues of $\mathbb{1}_4^{T_A}$ are

$$\left(\frac{1}{3}, \frac{1}{3}, \frac{1}{3}, \frac{1}{3}, \frac{4}{3}, \frac{4}{3}\right). \tag{C.3}$$

On the other hand, the eigenspectrum of $(|\psi\rangle\langle\psi|)^{T_A}$ in decreasing order is

$$\left(-\cos\alpha\sin\alpha, 0, 0, \sin^2\alpha, \sin\alpha\cos\alpha, \cos^2\alpha\right), \tag{C.4}$$

where $\alpha$ is the Schmidt angle of $|\psi\rangle \in \mathcal{H}_1^{\vee 3} \subset \mathcal{H}_1 \otimes \mathcal{H}_2$,

$$|\psi\rangle = \cos\alpha|n_1\rangle|m_1\rangle + \sin\alpha|n_1\rangle|m_2\rangle, \tag{C.5}$$

with $|n_k\rangle \in \mathcal{H}_1$ and $|m_k\rangle \in \mathcal{H}_2$ for $k = 1, 2$. We can observe that $(|\psi\rangle\langle\psi|)^{T_A}$, and then $\rho_S^{T_A}$, has at most one negative eigenvalue. The lowest eigenvalue $\Lambda_{\min}$ of

$$\rho_S^{T_A} = \left(x\mathbb{1}_4 + (1-4x)|\psi\rangle\langle\psi|\right)^{T_A} = x\mathbb{1}_4^{T_A} + (1-4x)(|\psi\rangle\langle\psi|)^{T_A}$$

is lower bounded by the sum of the minimum eigenvalue of each individual matrix

$$\frac{x}{3} - (1-4x)\cos\alpha\sin\alpha \leq \Lambda_{\min}. \tag{C.6}$$

In particular, the lower bound is minimized and achieved for $\alpha = \pi/4$ and

$$|\psi\rangle = \frac{1}{\sqrt{2}}\left(|+\rangle|++\rangle + |-\rangle|--\rangle\right), \tag{C.7}$$

which is the GHZ state. Therefore

$$\max_{U_S \in SU(4)} \mathcal{N}\left(U_S \rho_S U_S^{\dagger}\right) = \max\left(0, 1 - \frac{14x}{3}\right), \tag{C.8}$$

and the state $\rho_S$ is equal, up to a rotation, to $\rho_S^{(ii)}$. The previous result also shows that, along the simplex edge with $\tau_2 = \tau_3 = \tau_4$, the boundary with $\mathcal{A}_{\text{sym}}$ is given by the eigenspectrum $(\tau_1, \tau_2, \tau_3, \tau_4) = (5, 3, 3, 3)/14$, denoted by the point $p_2$ in Fig. 6.

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
