# Peer review of "Maximum entanglement of mixed symmetric states under unitary transformations"

_SciPost Physics, doi:SciPost Phys. 15, 120 (2023)_

## Round 2 · Referee Report · Anonymous (Referee 1) · 2023-6-2

Strengths

  1. The problems considered by the paper are completely natural, and I'm honestly surprised they weren't considered earlier. This work can be seen as a symmetric (bosonic) version of the paper "F. Verstraete, K. Audenaert, and B.D. Moor, Maximally entangled mixed states of two qubits, Phys. Rev. A, vol. 64, p. 012316, 2001" which I'm surprised wasn't written 10 or 15 years ago.

  2. The results obtained by the authors (in particular, Theorem 1 and Equation (13)) are deep. Neither the results nor their proofs are obvious, but they are simple-to-state and make use of.

Weaknesses

  1. The numerical experiments (i.e., Section 4) are less interesting and convincing than the results that came earlier in the paper. Maybe when mentioning the three-qubit results in the abstract and the introduction, the authors could mention up-front that only numerical results and conjectures are obtained in the three-qubit case.

Report

I believe that this paper will have high impact and will be of broad interest to the quantum information theory community. It clearly meets all of the journal's "general acceptance" criteria, and I believe that I meets "Expectations" #1 and #3 as well: Theorem 1 and Equation (13) are deep theoretical discoveries that are likely to launch new research projects.

Requested changes

  1. Typo near the bottom of page 5: " if it has more than one zero eigenvalues" should be " if it has more than one zero eigenvalue"

  2. Page 6, right before the start of Section 3: the authors talk about the maximally mixed symmetric state on 3 qubits. This state is indeed rank 4, so has 4 non-zero eigenvalues. The authors say that 2 of its 6 eigenvalues are 0... what are those 2? I could understand the statement "4 of its 8 eigenvalues are 0" since the symmetric subspace is a 4-dimensional subspace of (C^2)^{\otimes 3} = C^8... but I can't figure out where 6 dimensions are coming from. Maybe some clarification would help.

  3. In Equation (33), it took me too long to figure out what "s" is (I see now that it is a spin system parameter). Maybe remind the reader of what "s" is here, or replace it by N/2?

  4. In the abstract, the authors say "In particular, we determine the maximal radius of a ball of SAS states around the maximally mixed state in the symmetric sector, and the minimal radius of a ball that contains the set of SAS states." I read this sentence as saying that the authors computed these radii in all dimensions; something that Gurvits and Barnum did in the non-symmetric setting in 2002. However, the authors only computed these radii in the two-qubit case. I think that the abstract should be updated to make this clearer. I suggest a similar change on Page 4, right before the start of Section 2.

---

## Round 2 · Referee Report · Anonymous (Referee 2) · 2023-7-4

Report

This work is concerned with the characterization of entanglement in two or three-qubit states that live (fully) symmetric subspace of the underlying multipartite Hilbert space. The main reason to consider these particular situations is that in these systems partial transposition is an iff criterion for separability/entanglement. First, the authors study the two-qubit case and provide the following results:

  1. Derivation of the maximal value of negativity that can be achieved from a given symmetric mixed state by acting on it with an arbitrary global unitary operation that acts only on the symmetric subspace (Theorem 1). A direct corollary to Theorem 1 is a necessary and sufficient criterion for a state to be symmetric absolutely separable (SAS), where the latter stands for a state that is separable and remains separable under the action of any symmetric global unitary operation.

  2. Building on the above results, the authors then determine the minimal and maximal radii of the balls containing the set of symmetric absolutely separable two-qubit states.

  3. Then, the authors aim to determine the maximal amount of entanglement that can be obtained from a thermal state of a certain Hamiltonian by applying symmetric global unitary operations. In particular, values of the parameters that the Hamiltonians depends on are found for which the thermal states are symmetric absolutely separable.

The above results are then generalized to the symmetric three-qubit states for which partial transposition is also an iff criterion for separability. Since the three-qubit state is already too difficult to study analytically the authors employ some numerical methods. Based on the numerical search the authors pose a few conjectures such as that to obtain the maximal value of negativity over all operations from SU(4) it is enough to restrict to a certain four-parameter class of them. The authors also provide a sufficient condition for a symmetric three-qubit state to be SAS, formulated in terms of its eigenvalues.

I find this paper a solid piece of work. It asks interesting questions regarding characterization of entanglement in composite quantum systems. I particularly like the idea of studying the notion of absolutely separable state within quantum systems for which partial transposition detects all entangled states. The paper is also clearly written. I therefore recommend publication in Sci. Post.

Comments:

  1. Sometimes in the paper, (for instance in the abstract) the authors use the expression ‘permutation symmetry’ when talking about mixed states acting on the on the symmetric subspace of the entire Hilbert space. I would modify this expression somehow because in the literature permutationally invariant states are those that are invariant under a permutation of any pair of subsystems (see e.g. https://arxiv.org/pdf/1302.4100.pdf), and, while for pure states they are the same as those belonging to the symmetric subspace, for mixed states permutationally invariant states form a superset of the symmetric ones. For instance, a projection onto the two-qubit singlet state is permutationally invariant in the above sense.

  2. Perhaps the authors could include also [K. Eckert et al., Ann. Phys. 299, 88 (2002)] in the list of references, where the observation that partial transposition is an iff criterion for separability was made for the first time.

---

## Round 3 · Referee Report · Anonymous (Referee 1) · 2023-7-17

Report

I recommend publication; the author's changes have addressed all of my concerns.

---

## Round 3 · Author Response

Dear Editor,
We thank the referees for their valuable reports and their time reading and reviewing our
manuscript. We also appreciate their recommendations for publication in Sci. Post. We have
added all their requested changes in the new version of the manuscript. We comment below their
observations point-by-point.

We hope we have clarified all concerns and adequately addressed all questions. We believe this
reviewed version of our manuscript is in a suitable form for publication.

Yours sincerely,
Eduardo Serrano-Ensástiga and John Martin

---

## Round 3 · List of Changes

Answers to Report 1

Weakness 1. The numerical experiments (i.e., Section 4) are less interesting and convincing than the results that came earlier in the paper. Maybe when mentioning the three-qubit results in the abstract and the introduction, the authors could mention up-front that only numerical results and conjectures are obtained in the three-qubit case.

We are following the referee’s recommendation. The new version includes changes to highlight this comment in the abstract and introduction (page 4, before Section 2).

  1. Typo near the bottom of page 5: ” if it has more than one zero eigenvalues” should be ” if it has more than one zero eigenvalue”

We have corrected this typo.

  1. Page 6, right before the start of Section 3: the authors talk about the maximally mixed symmetric state on 3 qubits. This state is indeed rank 4, so has 4 non-zero eigenvalues. The authors say that 2 of its 6 eigenvalues are 0... what are those 2? I could understand the statement ”4 of its 8 eigenvalues are 0” since the symmetric 2 subspace is a 4-dimensional subspace of (C^2)⊗3 = C^8... but I can’t figure out where 6 dimensions are coming from. Maybe some clarification would help.

In this part of the text, we are considering the symmetric 3-qubit state as an 2 × 3 system, ρ ∈ B(H_1^{∨3} ) ⊂ B(H_1 ⊗ H_1^{∨2} ). Consequently, the density operator has 6 eigenvalues. For clarity of the reader, we have added additional text in the discussion.

  1. In Equation (33), it took me too long to figure out what ”s” is (I see now that it is a spin system parameter). Maybe remind the reader of what ”s” is here, or replace it by N/2?

We have rephrased this paragraph and Eqs. (33) and (34) in terms of the variable N.

  1. In the abstract, the authors say ”In particular, we determine the maximal radius of a ball of SAS states around the maximally mixed state in the symmetric sector, and the minimal radius of a ball that contains the set of SAS states.” I read this sentence as saying that the authors computed these radii in all dimensions; something that Gurvits and Barnum did in the non-symmetric setting in 2002. However, the authors only computed these radii in the two-qubit case. I think that the abstract should be updated to make this clearer. I suggest a similar change on Page 4, right before the start of Section 2.

We agree with the observation of the referee. We have updated the text before Section 2 and the abstract to make crystal-clear that we studied only these radii in the two- and three-qubit case.

Answers to Report 2

  1. Sometimes in the paper, (for instance in the abstract) the authors use the expression ‘permutation symmetry’ when talking about mixed states acting on the on the symmetric subspace of the entire Hilbert space. I would modify this expression somehow because in the literature permutationally invariant states are those that are invariant under a permutation of any pair of subsystems (see e.g. https://arxiv.org/pdf/1302.4100.pdf), and, while for pure states they are the same as those belonging to the symmetric subspace, for mixed states permutationally invariant states form a superset of the symmetric ones. For instance, a projection onto the two-qubit singlet state is permutationally invariant in the above sense.

We thank the referee for this accurate observation. We have made the necessary changes to avoid the term ”permutation invariant state”. More specifically, we have slightly edited the text in the abstract and on pages 3 (last paragraph), 4 (second paragraph), and 6 (first sentence).

  1. Perhaps the authors could include also [K. Eckert et al., Ann. Phys. 299, 88 (2002)] in the list of references, where the observation that partial transposition is an iff criterion for separability was made for the first time.

We thank the referee for mentioning the reference. We have added it, as well as the Reference to Peres. The new citations are [12] and [14] on the manuscript.

---

## Editorial Decision

published